# How the “Olive Oil Polyphenols” Health Claim Depends on Anthracnose and Olive Fly on Fruits

**DOI:** 10.3390/foods13111734

**Published:** 2024-06-01

**Authors:** Fátima Peres, Cecília Gouveia, Conceição Vitorino, Helena Oliveira, Suzana Ferreira-Dias

**Affiliations:** 1Instituto Politécnico de Castelo Branco, Escola Superior Agrária, 6000-909 Castelo Branco, Portugal; fperes@ipcb.pt (F.P.); cgouveia@ipcb.pt (C.G.); mcvitorino@ipcb.pt (C.V.); 2LEAF—Linking Landscape, Environment, Agriculture and Food Research Center, Instituto Superior de Agronomia, Universidade de Lisboa, Tapada da Ajuda, 1349-017 Lisbon, Portugal; 3Laboratório de Estudos Técnicos, Instituto Superior de Agronomia, Universidade de Lisboa, Tapada da Ajuda, 1349-017 Lisbon, Portugal

**Keywords:** *Bactrocera oleae*, *Colletotrichum* fungi, Cobrançosa olive cultivar, Galega olive cultivar, hydroxytyrosol, oleacein, polyphenols, tyrosol

## Abstract

Olive anthracnose, caused by *Colletotrichum* fungi, and the olive fruit fly *Bactrocera olea* are, respectively, the most important fungal disease and pest affecting olive fruits worldwide, leading to detrimental effects on the yield and quality of fruits and olive oil. This study focuses on the content of hydroxytyrosol (HYT) and its derivatives (the “olive oil polyphenols” health claim) in olive oils extracted from fruits of ‘Galega Vulgar’ and ‘Cobrançosa’ cultivars, naturally affected by olive anthracnose and olive fly. The olives, with different damage levels, were harvested from organic rainfed orchards, located in the center of Portugal, at four harvest times over three years. Galega oils extracted from olives with a higher anthracnose and olive fly incidence showed no conformity for the extra virgin olive oil (EVOO) and virgin olive oil (VOO) categories, presenting high acidity and negative sensory notes accompanied by the disappearance of oleacein. Conversely, no sensory defects were observed in Cobrançosa oils, regardless of disease and pest incidence levels, and quality criteria were still in accordance with the EVOO category. The total HYT and tyrosol (TYR) content (>5 mg/20 g) allows for the use of the “olive oil polyphenols” health claim on the label of all the analyzed Cobrançosa olive oils.

## 1. Introduction

A Mediterranean diet and olive oil intake have been related to longevity and a reduced risk of morbidity and mortality [1]. Along with some other characteristics of the Mediterranean diet, olive oil is used as the main source of fat in Portugal and other Southern European countries. The main advantages of consuming olive oil were traditionally attributed to its high content in monounsaturated fatty acids [2,3]. However, it is now well established that these effects must also be ascribed to the presence of phenolic compounds with different biologic activities, which are probably interconnected, like antioxidant, anti-inflammatory, and anti-microbial activities. For some activities of specific olive oil phenolic compounds, the evidence is already strong enough to enable the legal use of health claims [4,5,6]. Moreover, some authors recommend that the inclusion of a label with the health claim based on the olive oil polyphenols’ content would be useful to effectively signal both the “highest quality” and the “healthiest” extra virgin olive oils [7]. The phenolic composition of virgin olive oil (VOO) can also provide important information on its quality because phenols have an important impact on organoleptic evaluation, since they are responsible for positive sensory attributes of bitterness and pungency [8,9,10].

The fulfilment of the health claim “Olive oil polyphenols”, which can only be declared if the “olive oil contains more than 5 mg of hydroxytyrosol and its derivatives per 20 g of oil” [11], is dependent on the characteristics of the extracted fruits. In fact, the presence of these specific phenolic compounds in contents that meet the requirements of the health claim is strongly dependent on the olive cultivar, ripening, agronomic practices and geographical location, post-harvest, olive oil extraction, and storage [12,13,14,15,16,17,18]. Moreover, like with other phenolic compounds present in VOO, those ascribed to this health claim will be oxidized along the olive oil’s shelf-life, with a subsequent decrease in its content to levels below the stated health claim [19]. In addition, fruit damage caused by pests and diseases can have a huge negative impact on olive oil quality, especially in sensory characteristics and acidity [20].

Olive fruit fly *Bactrocera oleae* (Rossi) (Diptera: Tephritidae) is the main pest attacking the fruits in olive orchards, as well as the most devastating insect pest in the Mediterranean region [21,22,23]. The insect, alone or associated with microorganisms, strongly affects the quality and quantity of olives and olive oil. Female *B. oleae* lay their eggs under the surface of olive fruits, and the hatched larvae feed and develop in the fruit mesocarp, in which they form galleries, destroying the fruits [24]. Wounds caused by *B. oleae* in the fruits during oviposition and larvae feeding can enhance fruit infection by bacteria and fungi, thus promoting their fast deterioration. Furthermore, *B. oleae* can act as a carrier of phytopathogenic fungi, such as *Colletotrichum* spp. [25], responsible for olive anthracnose, thereby increasing its spread in the olive orchard. The susceptibility of different olive cultivars to the pest *B. oleae* is variable and depends on the interaction and correlation of physical, chemical, and molecular factors that modulate the olive fly preferences in relation to fruits. Among these factors, the size and color of the fruits (physical), the cuticle waxes, the content of phenolic or volatile compounds (chemical), and different genes that are affected as a result of the infestation (molecular) can be highlighted as examples [22].

Olive anthracnose is the most damaging olive fruit disease in many countries, including Portugal [26,27]. At least 18 *Colletotrichum* species are described as affecting olive trees worldwide. Seven species are present in Portugal, including *C. nymphaeae*, the most prevalent (73%), followed by *C. godetiae* and *C. acutatum* (9% each) [28]. *Colletotrichum* fungi cause the rot and dropping of mature olive fruits, the chlorosis and necrosis of leaves, and the dieback of twigs and branches. Olive anthracnose epidemics are promoted by autumn rainfall, a high inoculum density, fruit ripeness, and also depend on the susceptibility of host cultivars [29]. Regarding the susceptibility/tolerance of cultivars, previous studies indicate that this depends on the species of *Colletotrichum* present in the pathosystem and also on the isolate [30].

Oil extracted from olives affected by *Colletotrichum* spp. have off-flavors associated with a musty defect and high acidity [20,27,31]. Moreover, other compounds like phenols are also affected by the presence of anthracnose in the fruits [20,27,32].

Olive anthracnose disease and olive fly, in a climate change context with increased unpredictability, namely concerning the occurrence of periods of high humidity and mild temperatures, are a concern for the sustainable management of olive cultivation, harvesting, and processing for producing high-quality EVOO [20]. So, from the point of view of phenolic compounds, the assessment of the fulfilment of the health claim is an important research issue. As far as we know, the combined effect of olive fly attack and anthracnose, naturally occurring in olive fruits, in the fulfilment of the health claim “Olive oil polyphenols”, in olive oils extracted from these fruits, has not been studied yet. Therefore, the aim of the present work was to study the effect of anthracnose disease, together with olive fly attack, on the quality and phenolic compounds related to the health claim of the VOO obtained. Moreover, this was a three-year study conducted in the same orchards and with two different cultivars, namely ‘Galega Vulgar’ (highly susceptible to olive fly and anthracnose) and Cobrançosa (less susceptible to olive fly and tolerant to anthracnose).

## 2. Materials and Methods

### 2.1. Olive Characterization

Portuguese olive fruits (*Olea europaea* ssp. *europaea* var. *europaea*) of ‘Cobrançosa’ and ‘Galega Vulgar’ used in this study were produced in a rainfed olive orchard, without the use of pesticides, in the Beira Baixa region (39°49′ N, 7°27′ W). The climate of this region is classified as Csa (Mediterranean hot summer climates) according to the Köppen climate classification. Figure 1 shows the total rainfall and average temperature in each month during the three years of study in this region.

Olive fruits were picked from the first fortnight of October to the second fortnight of November (4 harvest times), in 2019, 2020, and 2021. Their ripening indices (RI) were determined following the guidelines of International Olive Council (IOC) [33]. The percentages of fruits infected by *Colletotrichum* spp. and attacked by olive fly (*Bactrocera oleae*) were evaluated (Figure 2). The incidence percentages of olive fly resulted from the visual observation of the number of fruits with the presence of the insect or insect damage (oviposition puncture, egg, larva, pupa, and larval/adult exit holes) in a sample of 100 fruits. For anthracnose, fruits were considered affected by *Colletotrichum* spp. when typical symptoms of the disease appeared, like round and ocher or brown lesions, with a profuse production of orange masses of conidia, or fruit rot. Observations were carried out in triplicate on a total of 300 fruits for each sample.

### 2.2. Olive Oil Extraction

On the same day of harvesting, and based on previous optimized conditions [34], olive oils were extracted in a laboratory oil extraction system (Abencor analyzer; MC2 Ingenieria y Sistemas S.L., Seville, Spain) using a hammer mill equipped with a 4 mm sieve at 3000 rpm, a malaxator (27–30 °C, 30 min), and a centrifuge (3500 rpm, 1 min). After centrifugation, the olive oil was separated by settling in a graduated cylinder. Water traces in the oil were removed with anhydrous sodium sulfate, filtered through a cellulose filter, and stored in amber glass bottles at 4 °C.

### 2.3. Chemical and Sensory Characterization of Olive Oil

Chemical quality criteria, considered by the European Union, including the acidity value (free fatty acids, FFA %), peroxide value (PV), and UV specific absorbances (K_232_ and K_270_), were evaluated for each VOO sample, as well as the major fatty acids (C16:0, C18:0, C18:1, and C18:2) by NIR spectroscopy using a spectrometer (MPA Bruker, Berlin, Germany), equipped with the calibration model B-Olive oil (Bruker). For the values that were not in conformity with the category of EVOO, a verification was performed by the official methods (Commission Delegated Regulation (EU) 2022/2104 of 29 July 2022) [35]. Samples of olive oils were also sensory evaluated by a panel test from the Laboratório de Estudos Técnicos, ISA, Portugal, recognized by the IOC [36]. A profile sheet with continuous 10 cm unstructured scales for negative and positive attributes (fruity, bitter, and pungent) was used [37]. Chlorophyll pigments (CP) were evaluated following the IUPAC method proposed by Pokorný et al. (1995) [38]. Total phenols (TPC) were extracted by liquid microextraction and were evaluated by VIS spectroscopy (JASCO 7800, Tokyo, Japan) as previously described [20]. For the quantification of hydroxytyrosol (HYT) and tyrosol (TYR) and their derivatives present in the olive oil samples, the hydromethanolic extracts prepared to quantify total phenols were used. Because a fraction of phenolic compounds is linked to other molecules, it is necessary to perform an acid hydrolysis of the phenolic extracts of olive oil for their full quantification [39,40]. For the hydrolysis process, the method of Nenadis et al. (2018) [41] was followed: 2 mL tubes with lids (in triplicate), containing 0.5 mL of phenolic extract and 0.5 mL of sulfuric acid (1 M), were shaken in a vortex (Labinco L46; Labinco BV, Breda, The Netherlands) for 15 s, placed in a thermostatic bath at 80 °C for 2 h, and then cooled in an ice bath. The analysis of the phenolic compounds HYT and TYR and their derivatives was carried out according to Reboredo-Rodríguez et al. (2016) [42], on an Agilent chromatograph, series 1100 (Agilent Technologies, Santa Clara, CA, USA), coupled with a C18 Phenomenex Kinetex column (100 mm × 3 mm, 2.6 μm) and diode array detector (DAD) (Agilent 1100; Agilent Technologies, Santa Clara, CA, USA). A gradient elution was performed with an eluent of water/formic acid (99.5:0.5; *v*/*v*) as mobile phase A and acetonitrile as mobile phase B. The total run time was 13 min (increased by 5 min post-run). The gradient elution conditions were as follows: 0 min, 95% A; 3 min, 80% A; 4 min, 60% A; 5 min, 55% A; 9 min, 40% A; 10 min, 0% A; 12 min, 95% A; and 13 min, 95% A. Concentrations of HYT and TYR were calculated based on the calibration curves established for HYT (R^2^ = 0.999; 6 data-points) and TYR (R^2^ = 0.999; 6 data-points), according to the method of Reboredo-Rodríguez et al. (2016) [42]. The value for the health claim corresponds to the sum of the amounts of HYT and TYR present in mg/20 g VOO. The phenolic profile of the olive oils extracted from fruits with different anthracnose disease was also evaluated by the IOC method, as previously described [43].

### 2.4. Statistical Analysis

Data obtained on fruit damage and chemical and sensory characterizations of olive oil samples were treated using Statistica^TM^ software, version 7, from Statsoft, Tulsa, OK, USA. An ANOVA was performed on the results concerning HYT, TYR, and “olive oil polyphenols” health claim values (in each year for 4 harvest moments). A post hoc Tukey test was used (*p* ≤ 0.05). A principal component analysis (PCA) was performed on the global dataset considering chemical and sensory results for all VOO samples obtained from Galega Vulgar or Cobrançosa fruits presenting different levels of biological damage and different ripening stages along the three harvest years. PCA is a pattern-recognition technique that will allow us to represent the original multidimensional dataset in a space with smaller dimensions and, therefore, help us to characterize the samples and evaluate the presence of eventual groups, as well as identify the most important variables on sample characterization [44]. Since the health claim value corresponds to the content of HYT and TYR in 20 g of olive oil, the parameter “Heath Claim” was used as a supplementary variable in PCA, while HYT and TYR were retained as active variables. Moreover, the ripening index (RI) was also used as a supplementary variable. These supplementary variables were not used to build principal components, but would help to interpret the variability of the data. Therefore, each VOO sample would be characterized by 18 variables: anthracnose attack, fly attack on the fruits, 4 chemical quality parameters, 4 main fatty acids, 4 sensory attributes, chlorophyll pigments (CP), total phenol content (TPC), and HYT and TYR contents.

## 3. Results and Discussion

### 3.1. Quality Criteria of the Oils

Along the three years, different levels of anthracnose and olive fly attack were expected due to the unpredictability of the climate, mainly in terms of the rainfall that occurred in autumn in 2020 and 2021, promoting latent and symptomatic infections of anthracnose (Figure 1). Quality criteria (acidity, peroxide value, and UV absorbances, in combination with organoleptic assessment) are the main parameters that define olive oil categories [35]. Table 1 shows that only Galega oils obtained in 2020 and 2021, in the last harvest time (RI of 6.4 and 5.1, respectively), that were extracted from fruits with a high incidence of anthracnose (91%) and olive fly attacks (69 and 72%, respectively), presented higher acidity (1.52 and 2.12%), and consequently were not in accordance with the category of extra virgin olive oil (EVOO; ≤0.8% FFA) or virgin olive oil (VOO; ≤2.0% FFA). The differences observed for acidity in both olive oils, in spite of apparently similar percentages of fruit damage, might be explained by their different disease severities (% of affected fruit surface and mummified fruits) [32]. Therefore, the Galega olive oil sample, presenting 2.12% acidity, belongs to the category of “lampante VOO” (L) and cannot be consumed without being refined. Moreover, these two samples presented the sensory defect “musty”, ascribed to anthracnose disease, confirming their low quality. All the samples showed peroxide values (PV: 3–10 meqO_2_/kg) and UV absorbance values (K_232_: 1.38–1.76.; K_270_: 0.07–0.18) lower than the maximum values allowed for edible VOO, indicating that VOO oxidation did not occur to a large extent in Galega oils. Therefore, the effect of anthracnose and olive fly attack is mainly related to the increase in acidity resulting from oil hydrolysis and with the appearance of the musty defect.

Table 2 shows the RI and the incidence of damaged Cobrançosa fruits by anthracnose and olive fly and the respective quality criteria of the extracted oils. All Cobrançosa oils presented low acidities (≤0.45%) and oxidation parameters (PV, K_232_ and K_270_) below the maximum values allowed for EVOO. Moreover, no sensory defects were detected, even when the oils were extracted from ripened fruits affected by anthracnose and olive fly. Therefore, all Cobrançosa samples belong to the category of EVOO. Olive fly attack in Cobrançosa olives is represented mainly by oviposition punctures (Figure 2), whereas, in the case of Galega fruits, the presence of eggs, larvae, pupae, and larval/adult exit holes are common. The different susceptibility of Galega and Cobrançosa to pests and diseases has already been explained by the cuticle thickness, perimeter, and area of epidermal cells [45], but the chemical composition of the fruits, like phenols or waxes, are also relevant [46,47].

Table 3 and Table 4 present the composition of both Galega and Cobrançosa olive oils extracted along the three harvests, concerning the main fatty acids (palmitic, stearic, oleic, and linoleic acids), the total content of phenolic compounds (TPC), and chlorophyll pigments (CP). Concerning fatty acid composition, Cobrançosa VOOs are richer in linoleic acid (C18:2), which varied from 7.3 to 10.4% (average = 8.8%), than Galega VOOs (average = 5.2%; min = 4.7%; max = 6.4%). Conversely, Galega oils present higher content of oleic acid (C18:1), varying from 75.0 to 77.3% (average = 75.9%), than Cobrançosa oils, with values ranging from 70.0 to 73.1% and an average of 71.7%. According to EC Regulation No 432/2012 [11], the health claims for oleic and for monounsaturated/polyunsaturated fatty acids are fulfilled for both Galega and Cobrançosa VOOs.

The results are in accordance with other studies that showed that fruit fly infestation and anthracnose disease did not cause an essential change in the fatty acid composition of olive oil [46]. However, an increase in linoleic acid (C18:2), especially in Cobrançosa oils, is observed, as was already reported in laboratory trials with olive oils extracted from olives in contact with the disease agents for several days [20]. In general, Cobrançosa oils are richer in phenolic compounds (average = 803.3 mg GAE/kg oil) than Galega oils (average = 415.7 mg GAE/kg oil). A decrease in TPC is observed along with maturation and with the damage of fruits, mainly for Galega oils, since the Galega cultivar is more susceptible to anthracnose disease than Cobrançosa. The same trend is observed for the green pigments (CP). In 2021, greener VOO were obtained for Cobrançosa oils, as olives were in a lower RI than in the other years. The lower contents for both cultivars in this year may be explained by meteorological conditions (September shows higher rainfall values) (Figure 1).

### 3.2. “Olive Oil Polyphenols” Health Claim vs. Olive Damage by Olive Fly and Anthracnose Disease

Most of the identified phenolic compounds in olive oil belong to five different classes: phenolic acids (especially derivatives of benzoic and cinnamic acids), flavonoids (luteolin and apigenin), lignans (pinoresinol and acetoxypinoresinol), phenolic alcohols (hydroxytyrosol and tyrosol), and secoiridoids (derivative aglycones of oleuropein and ligstroside) [18]. The group of secoiridoids, as conjugated forms of hydroxytyrosol and tyrosol, has been widely studied due to their health benefits and represents the phenolic family with the highest concentration in olive oil [10,48]. The phenolic fraction of olive oil depends on several parameters related to the quality of the fruits. In fact, the cultivar and ripening stage, when considering sound fruits, are the main decisive factors affecting the contents of phenolic compounds in VOOs. In addition, according to Gómez-Caravaca et al., no clear correlation seems to exist between the percentages of fly attack and phenolic content [49]. Moreover, verbascoside, tyrosol, and hydroxytyrosol were the compounds that were most adversely affected by *B. oleae* infestation, but fly attack was significantly correlated with the weight of the fruits, but not with the phenolic compounds [50]. In fact, anthracnose disease causing more fruit damage than olive fly, and producing lipases that hydrolyze the acylglycerols, releasing free fatty acids, can promote more differences in olive oil characteristics than olive fly, as observed in laboratory context studies [20]. However, when dealing with fruits collected in natural environments, where disease and pest control was not carried out, usually clear correlations are very difficult to find, due to several uncontrolled factors affecting the characteristics of the olives and consequently the olive oil. Harvest time, and consequently, the maturation of olives, are crucial for the presence of phenolic compounds in olives and olive oils [51,52]. In the case of the early ripening of susceptible cultivars to anthracnose disease, the destruction of the fruits will be more severe, the contact with mummified fruits will be longer, and a change in the phenolic profile is expected.

During the three harvests, the evolution of the phenolic compounds responsible for the olive oil polyphenols health claim (hydroxytyrosol + tyrosol) was evaluated, as well as that of the total phenolic compounds (Table 3 and Table 4) in the Galega and Cobrançosa olive oils under study. Table 5 shows the results of the evaluated parameters, as well as the value for the health claim. In the three years, the maximum and minimum levels of hydroxytyrosol in Cobrançosa olive oils were, respectively, 281.9 and 126.8 mg kg^−1^, while those of tyrosol were 251.0 and 132.9 mg kg^−1^. Comparatively, in Galega olive oils, the hydroxytyrosol varied between 2.76 and 189.71 mg kg^−1^, and tyrosol varied from 11.6 to 114.9 mg kg^−1^.

The results do not show a clear correlation between the percentage of anthracnose incidence and the phenolic compounds of the olive oils (Table 1 and Table 2). However, the concentration of HYT + TYR (health claim) is greater than 5 mg/20 g, for all Cobrançosa olive oils, throughout ripening in the three harvests and regardless of anthracnose and olive fly incidence (Table 5). On the contrary, in the 2020 and 2021 harvest years, none of the samples of Galega olive oil fulfilled the “Olive oil polyphenols” health claim, even at the beginning of ripening and at low disease incidence levels. In the 2019 harvest year, only Galega olive oils at the first harvest time (at the beginning of ripening, RI = 3.1) comply with the health claim requirement (Table 5). This can be also observed in olive oils extracted from olives that were infected with anthracnose but asymptomatic [53].

### 3.3. Phenolic Profiles of Olive Oils with Different Incidences of Anthracnose Disease

Regarding the main compounds mentioned in the literature, oleacein (HTY-EDA or 3,4-DHPEA-EDA) is the main phenolic compound present in virgin olive oil. Figure 3 shows the effect of anthracnose disease in the phenolic profile of Galega oils at two ripening moments in the 2020 harvest. The ripening stage is critical for the presence of several phenolic compounds [15,51]. Previous results on the phenolic profiles of VOOs extracted from fruits, performed by Peres et al. (2016) [51,52], with the same cultivars, showed a decrease in total phenolic content (TPC) beyond RI = 2–3, when healthy fruits were used in the extraction process, but representing a less than 30% decrease. However, this decrease is particularly important in the presence of pests and diseases. In the present study, oleacein and oleocanthal (p-HPEA-EDA) almost disappeared when anthracnose disease was present in 91% of the fruits, corresponding to a disease severity based on the affected surface of the fruits between 3 and 5 (3: 50–75%, 4: 100%, 5: mummified fruit) [32]. Oleacein decreased by 99%, from 242.54 to 2.5 mg kg^−1^, in the 2020 harvest for Galega oils (Figure 3).

The damage of Cobrançosa fruits was not assessed for the same ripening index of Galega fruits, since this last one is an early ripening cultivar. However, for Cobrançosa olive oils obtained at two ripening indices (2 and 4), the effect of the presence of olive fly and anthracnose was much smaller than for the Galega cultivar. A smaller decrease in the main chromatographic peaks was observed, but the main phenolic compounds did not disappear (Figure 4).

Although olive fly was already present at the beginning of ripening (RI = 2–3), when the first fruits were collected, the phenolic profiles are very similar to those of the olive oils from healthy fruits [51]. In fact, other authors [54] did not find any difference in the oleacein content between oils obtained from non-infested and infested fruits by *B. oleae*, either for the Picual or Hojiblanca cultivars.

### 3.4. Multivariate Characterizations of the Olive Oils

To assess the effect of both anthracnose and olive fly attack on the quality of the VOO, a PCA was performed on the global data, i.e., on a matrix with 24 samples (lines), defined by 18 active variables and 2 supplementary variables, namely the health claim “Olive oil polyphenols” (Claim) and the ripening index (RI). Through a PCA, it was possible to reduce the original 18-dimension space to a plane defined by the first two principal components (Factor 1 and Factor 2 of Figure 5). These first two factors have eigenvalues of 9.89 and 3.11, corresponding to a variance of 54.9 and 13.0%, respectively. Therefore, the plane defined by Factors 1 and 2 explains 72.2% of the variance and, therefore, of the information explained by the original dataset in an 18-dimension hyperspace. Figure 5a shows that the first principal component (Factor 1) is highly correlated with the positive sensory attributes of VOO, mainly bitterness and pungency, and with total phenolic compounds (TPH) and the contents of TYR and HYT. As expected, the health claim is highly correlated with TPH, TYR, and HYT. Therefore, Factor 1 can be identified as the “Olive oil polyphenols health claim” axis, since the parameters related to this claim are highly correlated with this axis, increasing along its negative side. Along this axis, towards the negative sense, the plotted samples present increasing quality and health claim values (Figure 5b). Cobrançosa VOO samples always have higher health claim values, are more pungent and bitter, and present higher TPC contents than the Galega VOO from fruits harvested on the same dates (Figure 5b).

Figure 5a shows that the anthracnose disease is highly correlated with the acidity increase (FFA) and the negative sensory attribute of being musty in VOOs. Olive fly attack is also related to these defects. This confirms that fly attack facilitates the infection of the fruits by *Colletotrichum* spp. Moreover, the ripening index (RI), considered an illustrative variable, shows to be highly correlated with the olive fruit susceptibility to anthracnose, confirming that ripened fruits are rather prone to this fungal disease, which will be responsible for the bad quality of their oil. In fact, the lowest quality VOO are the samples G42 and G43 of Galega Vulgar, harvested in 2020 and 2021, by the end of November (t4), as observed in Figure 5b. G42 and G43 samples were extracted from fruits with a high incidence of anthracnose (91%) and olive fly attack (69 and 72%, respectively), resulting in VOOs with higher acidity (1.52 and 2.12%) and musty sensory defects. The content of phenolic compounds considerably decreased to 87.05 and 137.9 mg GAE/kg, corresponding to 13 and 39% of the level in VOOs extracted from the fruits at the early ripening stage (early October). Consequently, these samples have very low values for the health claim: 0.29 and 0.42 mg/20 g VOO.

Figure 5b suggests a separation of Cobrançosa and Galega Vulgar VOOs into two groups, according to the cultivar. Along the second axis (Factor 2 or principal component 2), the content of chlorophyll pigments (CP) and oleic acid (C18:1) increase in the positive sense (Figure 5a). Galega VOOs are, in general, richer in green pigments and oleic acid than Cobrançosa VOOs (Figure 5b). However, a decrease in CP is observed for both cultivars along ripening. The reddish coloration intensity that is usually correlated to the proportion of symptomatic fruit with anthracnose [53] was only observed in the G43 sample.

From PCA, it is possible to conclude that the attack of the fruits by *Colletotrichum* spp. and/or olive fly has a more severe effect on the quality of Galega Vulgar VOO than on Cobrançosa VOO. PCA also confirms that early harvesting is an adequate strategy to avoid high levels of anthracnose and fly attack, preserving the quality of VOOs, for both cultivars, but is mandatory for Galega olives, especially under the changing climate.

## 4. Conclusions

This is the first study on the effect of olive anthracnose and olive fly attack on the fruits on the decrease in the contents of hydroxytyrosol (HYT) and its derivatives, related to the health claim “Olive oil polyphenols”.

The results obtained in this study showed that, even with high levels of damaged fruits by anthracnose and/or olive fly, almost all the extracted VOOs can still be classified as EVOO, and that health claims for oleic and for monounsaturated/polyunsaturated fatty acids are fulfilled.

Oils from the ‘Galega’ cultivar, considered very susceptible to *Colletotrichum* fungi and olive fruit fly, only fulfilled the health claim related to “olive oil polyphenols” in the first year of observations (2019) and at the first two harvest moments. This is a very important finding for this cultivar because these virgin oils are very resistant to oxidation and very appreciated by Portuguese consumers. For the ‘Cobrançosa’ cultivar, considered moderately susceptible to *Colletotrichum* fungi and olive fruit fly, all the oils studied fulfilled the health claim for polyphenols, even the ones extracted from fruits with higher fly/anthracnose damage levels. However, *Colletotrichum* infection in fruits, with concomitant infestation by *B. oleae*, may compromise the use of Cobrançosa oils in award-winning olive oil blends, mainly due to a decrease in the intensity of the pungent and bitter positive attributes, as observed in 2020 and 2021 harvests.

The importance of early ripening to preserve phenolic compounds in olive oils is again shown in the present study. However, the early harvesting time may not be sufficient for Galega oils to fulfil the health claim “olive oil polyphenols”. Therefore, the impossibility to fulfil this health claim can bring the need to modulate the very best agronomic and technological practices to control olive fruit fly and anthracnose for this cultivar. Regarding the quality of the olive oil obtained, this study shows that the ‘Cobrançosa’ cultivar can deal better with olive fly and *Colletotrichum* spp. attacks than ‘Galega Vulgar’ can.

## Figures and Tables

**Figure 1 foods-13-01734-f001:**
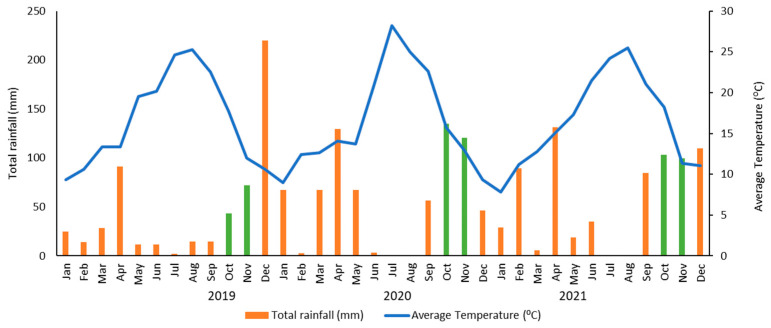
Monthly total rainfall (bars) and average temperature (solid line) in 2019, 2020, and 2021 in the Beira Baixa region (39°49′ N, 7°27′ W), Portugal. The green bars correspond to the months in which the samples were collected.

**Figure 2 foods-13-01734-f002:**
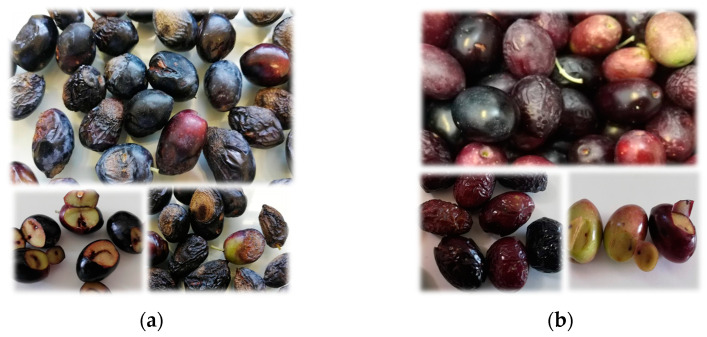
Olive fruits, at the same ripening index, damaged by *Colletotrichum* fungi (anthracnose) and *Bactrocera olea* (olive fruit fly): (**a**) ‘Galega Vulgar’ cultivar and (**b**) ‘Cobrançosa’ cultivar. Anthracnose symptoms ranged from wrinkled fruits to fruits with sunken lesions, with some exhibiting spore masses, to rotten fruits; olive fruit fly damage was mainly revealed by larval feeding tunnels in olive mesocarp.

**Figure 3 foods-13-01734-f003:**
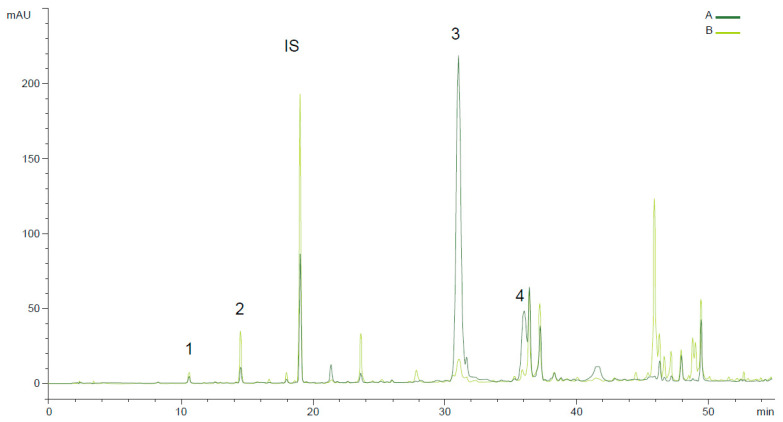
Phenolic profile (RP-HPLC-UV 280 nm) of Galega olive oil obtained with olives at two ripening moments and with different severities of anthracnose disease; (A) RI = 3 and anthracnose = 5%; (B) RI = 6 and anthracnose—91%. 1—HYT, 2—TYR, IS—internal standard (syringic acid), 3—oleacein, and 4—oleocanthal.

**Figure 4 foods-13-01734-f004:**
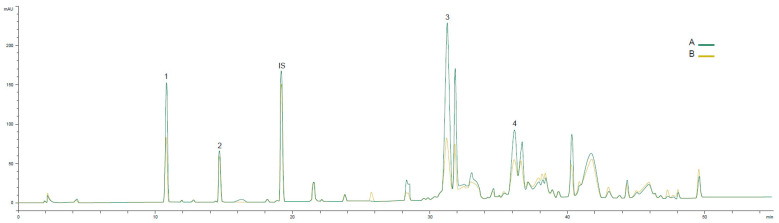
Phenolic profile (RP-HPLC-UV 280 nm) of Cobrançosa olive oil obtained with olives at two ripening moments and with different anthracnose disease severities; (A) RI = 2 and anthracnose −30%; (B) RI = 4 and anthracnose = 64%. 1—HYT, 2—TYR, IS—internal standard (syringic acid), 3—oleacein, and 4—oleocanthal.

**Figure 5 foods-13-01734-f005:**
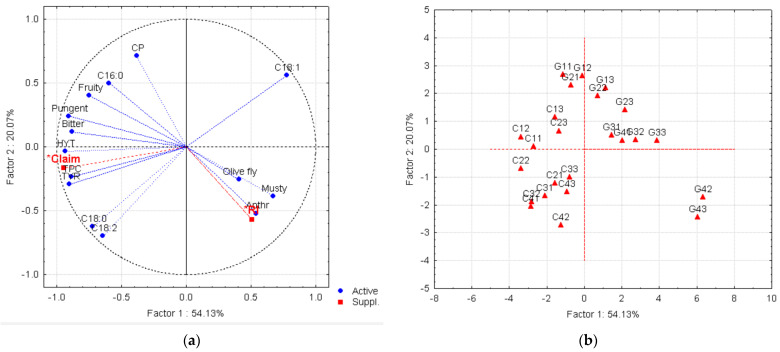
Plot of (**a**) the loadings of the original variables and (**b**) of the VOO samples on the plane defined by principal components 1 (Factor 1) and 2 (Factor 2) (C: Cobrançosa VOO samples; G: Galega VOO samples; the subscripts of the samples, ty, correspond to the harvest time (t = 1, …, 4) and year (y = 1, …, 3, where 1 = 2019; 2 = 2020; 3 = 2021); the abbreviations of the variables are the same as those used in Table 1, Table 2, Table 3, Table 4 and Table 5). “Olive oil polyphenols” health claim (*Claim) and ripening index (*RI) are supplementary variables.

**Table 1 foods-13-01734-t001:** Quality criteria: acidity (FFA), peroxide value (PV), UV absorbances (K_270_, K_232_), and sensory attributes (Fruity = green fruity; Pung. = pungent) for Galega oils extracted from olives with different damage caused by olive fly (Fly, %) and anthracnose disease (Ant., %), and ripening index (RI). The values of chemical parameters are the average of three replicates ± standard deviations. Olive oil category (Cat.): EVOO—extra virgin olive oil; VOO—virgin olive oil; and L—lampante olive oil.

Year	Ant. (%)	Fly (%)	RI	FFA (%)	PV (meqO_2_ kg^−1^)	K_232_	K_270_	Fruity	Bitter	Pung.	Musty	Cat.
2019	9	47	3.1	0.38 ± 0.00	8.60 ± 0.24	1.65 ± 0.02	0.17 ± 0.01	6.1	7.7	7.7	0	EVOO
	5	57	3.1	0.37 ± 0.00	9.92 ± 0.13	1.76 ± 0.01	0.13 ± 0.01	5.2	7.3	7.8	0	EVOO
	51	82	4.1	0.50 ± 0.00	7.44 ± 0.08	1.57 ± 0.01	0.13 ± 0.01	4.6	4.9	4.6	0	EVOO
	70	35	5.8	0.64 ± 0.00	6.53 ± 0.07	1.50 ± 0.01	0.12 ± 0.01	3.2	6.0	5.0	0	EVOO
2020	3	44	1.8	0.34 ± 0.00	9.95 ± 0.08	1.69 ± 0.02	0.18 ± 0.01	6.1	9.1	8.9	0	EVOO
	5	16	2.9	0.23 ± 0.01	8.61 ± 0.25	1.58 ± 0.03	0.12 ± 0.00	4.6	7.3	5.2	0	EVOO
	25	42	4.8	0.23 ± 0.00	4.30 ± 0.18	1.50 ± 0.02	0.10 ± 0.01	3.4	4.6	2.8	0	EVOO
	91	69	6.4	1.52 ± 0.01	3.18 ± 0.38	1.46 ± 0.01	0.10 ± 0.01	0.1	0	0	1.2	VOO
2021	0	15	2.1	0.32 ± 0.00	9.02 ± 0.04	1.71 ± 0.01	0.13 ± 0.01	3.9	2.8	4.3	0	EVOO
	0	41	3.5	0.33 ± 0.00	6.50 ± 0.03	1.53 ± 0.02	0.11 ± 0.01	3.4	2.2	3.7	0	EVOO
	3	59	3.8	0.42 ± 0.00	4.23 ± 0.15	1.38 ± 0.01	0.07 ± 0.01	1.6	0	0.6	0	EVOO
	91	72	5.1	2.12 ± 0.01	4.33 ± 0.10	1.56 ± 0.01	0.13 ± 0.01	0.1	0	0	1.9	L

**Table 2 foods-13-01734-t002:** Quality criteria: acidity, peroxide value (PV), UV absorbances (K_270_, K_232_), and sensory attributes (Fruity = green fruity; Pung. = pungent) for Cobrançosa oils extracted from olives with different damage caused by olive fly (Fly, %) and anthracnose disease (Ant., %), and ripening index (RI). The values of chemical parameters are the averages of three replicates ± standard deviations. Olive oil category (Cat.): EVOO—extra virgin olive oil.

Year	Ant. (%)	Fly (%)	RI	FFA (%)	PV meqO_2_ kg^−1^	K_232_	K_270_	Fruity	Bitter	Pung.	Cat.
2019	0	36	1.6	0.26 ± 0.00	15.23 ± 0.12	2.05 ± 0.01	0.19 ± 0.01	7.0	8.1	7.9	EVOO
	30	83	3.8	0.30 ± 0.00	11.63 ± 0.15	1.97 ± 0.01	0.16 ± 0.01	7.0	7.3	7.6	EVOO
	44	63	3.5	0.45 ± 0.00	10.93 ± 0.07	1.85 ± 0.01	0.14 ± 0.00	5.2	6.8	6.0	EVOO
	64	36	5.2	0.42 ± 0.00	11.43 ± 0.13	1.86 ± 0.01	0.12 ± 0.00	5.9	8.7	6.2	EVOO
2020	6	29	2.4	0.34 ± 0.001	15.51 ± 0.16	1.96 ± 0.01	0.19 ± 0.01	4.3	9.4	9.8	EVOO
	4	14	3.4	0.28 ± 0.00	13.06 ± 0.11	2.02 ± 0.01	0.17 ± 0.01	4.3	9,1	9.5	EVOO
	4	26	3.9	0.28 ± 0.00	11.20 ± 0.10	1.93 ± 0.01	0.10 ± 0.00	5.1	8.1	7.3	EVOO
	12	40	4.7	0.22 ± 0.00	7.73 ± 0.13	1.78 ± 0.01	0.09 ± 0.00	2.7	7.6	6.7	EVOO
2021	0	27	1.0	0.32 ± 0.00	12.55 ± 2.17	1.87 ± 0.02	0.18 ± 0.04	6.5	6.1	7.2	EVOO
	0	46	1.9	0.34 ± 0.00	11.21 ± 0.09	1.77 ± 0.02	0.15 ± 0.01	6.3	5.7	7.1	EVOO
	24	73	2.7	0.34 ± 0.00	10.91 ± 0.15	1.73 ± 0.01	0.14 ± 0.01	5.5	5.6	6.5	EVOO
	9	28	3.8	0.39 ± 0.00	10.13 ± 0.77	1.71 ± 0.02	0.10 ± 0.03	4.0	4.6	4.5	EVOO

**Table 3 foods-13-01734-t003:** Galega olive oil composition: main fatty acids, total phenols (TPC), and chlorophyl pigments (CP) for the harvest times reported in Table 1. The values are the average of three replicates ± standard deviations.

Year	C16:0 (%)	C18:0 (%)	C18:2 (%)	C18:1 (%)	TPC mg GAE/kg	CP mg/kg
2019	15.76 ± 0.04	2.52 ± 0.04	4.94 ± 0.07	74.98 ± 0.14	626.41 ± 27.21	65.49 ± 0.04
	15.33 ± 0.04	2.67 ± 0.01	4.96 ± 0.07	75.04 ± 0.07	504.82 ± 12.68	50.60 ± 0.05
	14.52 ± 0.03	2.59 ± 0.01	5.22 ± 0.03	75.29 ± 0.05	458.49 ± 2.20	28.44 ± 0.25
	14.04 ± 0.01	2.39 ± 0.01	4.95 ± 0.02	76.54 ± 0.02	468.29 ± 10.72	1.72 ± 0.02
2020	13.95 ± 0.02	2.22 ± 0.02	4.90 ± 0.02	76.49 ± 0.04	681.43 ± 4.16	61.24 ± 0.01
	14.25 ± 0.06	2.42 ± 0.05	4.76 ± 0.08	75.90 ± 0.14	539.03 ± 9.85	34.43 ± 0.02
	13.33 ± 0.07	2.44 ± 0.02	5.32 ± 0.07	76.16 ± 0.14	337.09 ± 12.67	4.96 ± 0.01
	12.60 ± 0.04	2.53 ± 0.03	4.70 ± 0.05	77.28 ± 0.07	87.05 ± 4.71	5.37 ± 0.01
2021	14.35 ± 0.01	2.30 ± 0.03	5.42 ± 0.04	75.18 ± 0.06	352.22 ± 6.76	64.88 ± 0.59
	14.08 ± 0.01	2.22 ± 0.01	5.54 ± 0.03	75.60 ± 0.03	274.85 ± 5.13	30.58 ± 0.10
	13.58 ± 0.04	2.30 ± 0.04	5.55 ± 0.02	75.75 ± 0.06	210.15 ± 5.73	8.55 ± 0.02
	13.27 ± 0.05	2.42 ± 0.04	6.43 ± 0.03	75.17 ± 0.07	137.90 ± 2.77	2.50 ± 0.02

**Table 4 foods-13-01734-t004:** Cobrançosa olive oil composition: main fatty acids, total phenols (TPC), and chlorophyl pigments (CP) for the harvest times reported in Table 2. The values are the averages of three repetitions ± standard deviations.

Year	C16:0 (%)	C18:0 (%)	C18:2 (%)	C18:1 (%)	TPC mg GAE/kg	CP mg/kg
2019	14.78 ± 0.02	3.12 ± 0.03	7.71 ± 0.02	72.42 ± 0.03	762.33 ± 16.23	28.39 ± 0.01
	14.28 ± 0.03	3.13 ± 0.01	8.82 ± 0.03	71.48 ± 0.01	718.72 ± 2.53	21.21 ± 0.04
	14.66 ± 0.02	3.11 ± 0.03	9.03 ± 0.03	71.57 ± 0.03	1019.05 ± 11.49	16.29 ± 0.01
	14.25 ± 0.03	3.09 ± 0.03	8.88 ± 0.07	71.81 ± 0.02	1233.62 ± 12.13	3.57 ± 0.06
2020	14.81 ± 0.01	3.09 ± 0.02	7.25 ± 0.05	72.33 ± 0.06	1081.71 ± 12.13	60.95 ± 0.01
	14.32 ± 0.05	3.20 ± 0.02	9.78 ± 0.03	70.25 ± 0.03	904.14 ± 5.58	49.65 ± 0.01
	14.08 ± 0.05	3.39 ± 0.03	10.39 ± 0.06	69.96 ± 0.08	911.16 ± 14.20	7.82 ± 0.02
	13.66 ± 0.04	3.56 ± 0.04	10.26 ± 0.01	70.15 ± 0.07	710.95 ± 27.48	1.19 ± 0.01
2021	13.95 ± 0.11	2.89 ± 0.03	7.29 ± 0.05	72.93 ± 0.07	501.17 ± 7.13	79.94 ± 0.22
	13.89 ± 0.29	2.91 ± 0.03	7.99 ± 0.13	73.05 ± 0.03	526.18 ± 92.01	71.65 ± 0.05
	13.61 ± 0.04	2.96 ± 0.05	8.80 ± 0.01	72.64 ± 0.04	607.72 ± 8.79	33.39 ± 0.02
	13.60 ± 0.05	3.10 ± 0.01	9.39 ± 0.02	71.38 ± 0.10	662.80 ± 1.96	16.32 ± 0.01

**Table 5 foods-13-01734-t005:** Hydroxytyrosol (HYT) and tyrosol (TYR) and “Olive oil polyphenols” health claim values for Cobrançosa and Galega oils obtained in the three harvest years and along with maturation, corresponding to the samples reported in Table 1 and Table 2. In each column, different letters indicate that the results are significantly different at *p* ≤ 0.05 (Tukey test) in each year. The values are the averages of three replicates ± standard deviations.

Cobrançosa Virgin Olive Oils				Galega Virgin Olive Oils		
Year	HYT (mg/kg)	TYR (mg/kg)	Health Claim (mg/20 g)	HYT (mg/kg)	TYR (mg/kg)	Health Claim (mg/20 g)
2019	199.92 ± 14.58 ^b^	228.09 ± 4.21 ^b^	8.56 ± 0.13 ^b^	189.71 ± 1.09 ^a^	114.94 ± 5.98 ^a^	6.09 ± 0.14 ^a^
	126.87 ± 3.39 ^c^	199.91 ± 1.61 ^c^	6.54 ± 0.04 ^c^	146.57 ± 10.03 ^b^	107.02 ± 9.87 ^a^	5.07 ± 0.03 ^b^
	223.43 ± 5.46 ^b^	224.90 ± 1.27 ^b^	8.97 ± 0.11 ^b^	120.29 ± 1.19 ^c^	85.94 ± 1.09 ^b^	4.12 ± 0.05 ^c^
	281.93 ± 9.91 ^a^	251.04 ± 1.84 ^a^	10.66 ± 0.20 ^a^	109.21 ± 3.27 ^d^	69.23 ± 0.13 ^c^	3.6 ± 0.08 ^d^
2020	261.66 ± 35.13 ^a^	207.52 ± 3.51 ^a^	9.38 ± 0.77 ^a^	143.79 ± 12.02 ^a^	60.36 ± 4.70 ^a^	4.08 ± 0.16 ^a^
	216.06 ± 5.04 ^b^	201.29 ± 26.93 ^a^	8.35 ± 0.63 ^b^	108.41 ± 8.71 ^b^	65.61 ± 5.03 ^a^	3.48 ± 0.27 ^b^
	213.85 ± 5.67 ^b^	169.39 ± 5.42 ^b^	7.66 ± 0.11 ^b^	56.39 ± 4.66 ^c^	40.62 ± 2.31 ^b^	1.94 ± 0.14 ^c^
	138.08 ± 2.00 ^c^	132.93 ± 4.05 ^c^	5.42 ± 0.07 ^c^	2.76 ± 0.97 ^d^	11.57 ± 2.62 ^c^	0.29 ± 0.09 ^d^
2021	201.17 ± 2.25 ^a^	162.52 ± 0.86 ^a^	7.27 ± 0.06 ^a^	125.12 ± 3.03 ^a^	80.52 ± 0.73 ^a^	4.1 ± 0.45 ^a^
	196.54 ± 11.71 ^a^	161.07 ± 0.89 ^a^	7.15 ± 0.25 ^a^	109.20 ± 0.53 ^b^	56.36 ± 0.31 ^b^	3.31 ± 0.02 ^b^
	195.98 ± 8.94 ^a^	165.94 ± 1.45 ^a^	7.24 ± 0.15 ^a^	41.20 ± 0.17 ^c^	30.82 ± 2.11 ^c^	1.44 ± 0.04 ^c^
	209.08 ± 0.10 ^a^	146.96 ± 0.56 ^b^	7.12 ± 0.01 ^a^	3.51 ± 0.98 ^d^	17.34 ± 0.24 ^d^	0.42 ± 0.02 ^d^

## Data Availability

The original contributions presented in the study are included in the article, further inquiries can be directed to the corresponding author.

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
