# Peer review of "How the “Olive Oil Polyphenols” Health Claim Depends on Anthracnose and Olive Fly on Fruits"

_foods, 2024, doi:10.3390/foods13111734_

Round 1

Reviewer 1 Report

Comments and Suggestions for Authors

Manuscript entitled “How “olive oil polyphenols” health claim depends on anthracnose and olive fly on fruits?” is clear, relevant to the field and well-structured. References cited in the paper are relevant and recent publications. The aim of the paper is interesting, scientific papers have so far been poorly published on this topic. Anthracnose and olive fly are common pests in olive orchards, so, from a practical point of view, it is important to examine their influence on the content of polyphenols. The methods are reported in sufficient detail to allow for its replicability and/or reproducibility. Results are clearly presented and the obtained conclusions derive from the results. However, authors should pay attention to the following sections:

Line 272: very difficult to find

Table 2: 2019 (Year)

4. Conclusions section is written too extensively, conclusions should be more concise.

Author Response

Dear Colleague,

We would like to thank you for taking the time to review our manuscript “How “olive oil polyphenols” health claim depends on anthracnose and olive fly on fruits”, submitted to Foods and for your positive evaluation and suggestions to improve it. The answers to your questions and comments are presented below and the corrections are highlighted in yellow in the revised manuscript.

Best regards,

Suzana Ferreira-Dias

 Author's Reply to the Review Report (Reviewer 1)

Comments and Suggestions for Authors

Manuscript entitled “How “olive oil polyphenols” health claim depends on anthracnose and olive fly on fruits?” is clear, relevant to the field and well-structured. References cited in the paper are relevant and recent publications. The aim of the paper is interesting, scientific papers have so far been poorly published on this topic. Anthracnose and olive fly are common pests in olive orchards, so, from a practical point of view, it is important to examine their influence on the content of polyphenols. The methods are reported in sufficient detail to allow for its replicability and/or reproducibility. Results are clearly presented and the obtained conclusions derive from the results. However, authors should pay attention to the following sections:

Line 272: very difficult to find.

Answer: corrected (line 289)

Table 2: 2019 (Year)

Answer: corrected.

  1. Conclusions section is written too extensively, conclusions should be more concise.

Answer: Thank you very much for the suggestion. The conclusions were shortened as follows:

“This is the first study on the effect of olive anthracnose and olive fly attack of the fruits on the decrease in the contents of hydroxytyrosol (HYT) and its derivatives, related to the health claim “Olive oil polyphenols”.

The results obtained in this study showed that, even with high levels of damaged fruits by anthracnose and/or olive fly, almost all the extracted VOO can still be classified as EVOO, and that health claims for oleic and for monounsaturated/polyunsaturated fatty acids are fulfilled.

Oils from ‘Galega’ cultivar, considered very susceptible to Colletotrichum fungi and olive fruit fly, only fulfilled the health claim related to “olive oil polyphenols” in the first year of observations (2019) and in the first two harvest moments. This is a very important finding for this cultivar because these virgin oils are very resistant to oxidation and very appreciated by the Portuguese consumers. For ‘Cobrançosa’ cultivar, considered moderate susceptible to Colletotrichum fungi and olive fruit fly, all the oils studied fulfilled the health claim for polyphenols, even the ones extracted from fruits with higher fly/anthracnose damages. However, Colletotrichum infection in fruits with concomitant infestation by B. oleae may compromise the use of Cobrançosa oils in award-winning olive oil blends, mainly due to a decrease in the intensity of the pungent and bitter positive attributes, as observed in 2020 and 2021 harvests.

The importance of early ripening to preserve phenolic compounds in olive oils is again shown in the present study. However, the early harvesting time may not be sufficient for Galega oils to fulfil the health claim “olive oil polyphenols”. Therefore, the impossibility to fulfil this health claim can bring the need to modulate the very best agronomic and technological practices to control olive fruit fly and anthracnose for this cultivar. Regarding the quality of the olive oil obtained, this study shows that the ‘Cobrançosa’ cultivar can deal better with olive fly and Colletotrichum spp. attacks than ‘Galega Vulgar’.”

Reviewer 2 Report

Comments and Suggestions for Authors

The study focuses on 15 the content of bioactive compounds, namely on hydroxytyrosol (HYT) and its derivatives (“olive oil 16 polyphenols” health claim) in olive oils extracted from fruits of 'Galega Vulgar' and 'Cobrançosa' 17 cultivars, naturally attacked by olive anthracnose and olive fly. The manuscript hypothesis was interesting, and it has good results for the literature. It can be interesting for readers. But the following factors should be taken into account;

The results section can be supported with more references.

The abstract should be reformed according to journal rules.

A graphical abstract can be formed.

Why were the harvesting years selected as 2019-2020-202? Can the three years reflect the hypothesis? 

Author Response

Dear Colleague,

We would like to thank you for taking the time to review our manuscript “How “olive oil polyphenols” health claim depends on anthracnose and olive fly on fruits”, submitted to Foods and for your positive evaluation and suggestions to improve it. The answers to your questions and comments are presented below and the corrections are highlighted in yellow in the revised manuscript.

Best regards,

Suzana Ferreira-Dias

Author's Reply to the Review Report (Reviewer 2)

 Comments and Suggestions for Authors

 The study focuses on the content of bioactive compounds, namely on hydroxytyrosol (HYT) and its derivatives (“olive oil polyphenols” health claim) in olive oils extracted from fruits of 'Galega Vulgar' and 'Cobrançosa' cultivars, naturally attacked by olive anthracnose and olive fly. The manuscript hypothesis was interesting, and it has good results for the literature. It can be interesting for readers. But the following factors should be taken into account;.

Question: The results section can be supported with more references.

Ans: You were right. Citations were added in line 206 (ref 35), line 214 (ref 32) and line 334 (ref 15 and 51).

Question: The abstract should be reformed according to journal rules.

Ans: The abstract was shortened to 200 words, according to journal rules, as follows.

“Olive anthracnose, caused by Colletotrichum fungi, and the olive fruit fly Bactrocera olea are, respectively, the most important fungal disease and pest affecting olive fruits worldwide, leading to detrimental effects on yield and quality of fruits and olive oil. This study focuses on the content of hydroxytyrosol (HYT) and its derivatives (“olive oil polyphenols” health claim) in olive oils extracted from fruits of 'Galega Vulgar' and 'Cobrançosa' cultivars, naturally affected by olive anthracnose and olive fly. The olives, with different damage levels, were harvested from organic rainfed orchards, located in the Center of Portugal, at four harvest times over three years. Galega oils extracted from olives with a higher anthracnose and olive fly incidence showed no conformity for the Extra Virgin Olive Oil (EVOO) and Virgin Olive Oil (VOO) categories, presenting high acidity and negative sensory notes accompanied by the disappearance of oleacein. Conversely, no sensory defects were observed in Cobrançosa oils, regardless the disease and pest incidence levels, and quality criteria were still in accordance with the EVOO category. The total HYT and tyrosol (TYR) content (> 5mg/20g) allows the use of the “olive oil polyphenols” health claim on the label of all the analyzed Cobrançosa olive oils.”

.Question: A graphical abstract can be formed.

Ans: A graphical abstract was added as suggested.

Question: Why were the harvesting years selected as 2019-2020-202? Can the three years reflect the hypothesis? 

Ans: This study was funded by a 3-year research project during 2019, 2020 and 2021. If these years are representative of the global situation in the region under study (Centre-interior of Portugal, close to Spain) is not possible to tell. It would be necessary to get funding to continue these studies along several years, which is not the case.

Reviewer 3 Report

Comments and Suggestions for Authors

Author Response

Dear Colleague,

We would like to thank you for taking the time to review our manuscript “How “olive oil polyphenols” health claim depends on anthracnose and olive fly on fruits”, submitted to Foods, and for your positive evaluation and suggestions to improve it. The answers to your questions and comments are presented below and the corrections are highlighted in yellow in the revised manuscript.

Best regards,

Suzana Ferreira-Dias

 Author's Reply to the Review Report (Reviewer 3)

 Comments and Suggestions for Authors

Question: Please add olive fly.

Ans: Keywords: the “olive fly” was added.

Question: Please illustrate the stages of infestation recorded with photos.

Ans: A figure (Fig. 2) was added with examples of Galega and Cobrançosa fruits infected by anthracnose and attacked by olive fly.

Question: Indicate the sensory evaluation method used.

Ans: The sensory evaluation method was described as follows:

Samples of olive oils were also sensory evaluated by a panel test from the Laboratório de Estudos Técnicos, ISA, Portugal, recognized by the IOC [35]. A profile sheet with continuous 10 cm unstructured scales for negative and positive attributes (fruity, bitter, and pungent) was used [37].

Question: Show the calculation methods of health claim and give references.

Ans: Calculation methods of olive oil polyphenols health claim was added as follows:

“Concentrations of HYT and TYR were calculated based on the calibration curves established for HYT (R2= 0.999; 6 data-points) and TYR (R2 = 0.999; 6 data-points), according to the method of Reboredo-Rodríguez et al (2016) [42].

Question: Following your harvesting and extraction procedure, please indicate the origin of this defect in your sample!

.

Ans: The sensory defect “musty” was ascribed to anthracnose disease (added to the text, lines 216-217).

Question: In your case, two important attributes need to be evaluated: fruitiness and Grabby.

Ans: Tables 1 and 2: the green fruity was added. However, “grubby” defect was not quantified because it was considered in the group of “other negative attributes”.

Question:  Thank you for explaining this situation. Same degree of attack without the same effect! Why or explain the reason?

Ans: The explanation of the differences observed for acidity in both olive oils, in spite of apparently similar percentage of fruit damage, was as follows (lines 2111-214):

“The differences observed for acidity in both olive oils, in spite of apparently similar percentage of fruit damage, might be explained by different disease severity (% of affected sur-face fruit and mummified fruits) [32].”

Question: The difference in maturity stage between the two varieties poses a critical question. Indeed, in this kind of comparative study, the same conditions, in particular the same stage of maturity, is important for a correct assessment of varietal performance.

 Ans: We agree with you, but we decided to fix the harvest date for both cultivars knowing that it would results in different ripening indexes.

Question: Please indicate the main stage of attack at each stage of maturity: (oviposition puncture, egg, larva, pupa and larval/adult exit holes).

Ans: As indicated in the manuscript, the incidence percentages of olive fly resulted from the visual observation of the number of fruits with the presence of the insect or insect damages (oviposition puncture, egg, larva, pupa and larval/adult exit holes) in a sample of 100 fruits. Thus, we do not have the detailed information you requested.  

Question: Table 5: These results are very much affected by the stage of ripening, making it difficult to assess the impact of infestation. Please take these observations into account when interpreting your results.

Ans: We agree with you, and this issue was previously mentioned in the first version of this manuscript, as follows:

“The ripening stage is critical for the presence of several phenolic compounds. Previous results on phenolic profile of VOO extracted from fruits, performed by Peres et al (2016) [49,50], with the same cultivars, showed a decrease in total phenolic content (TPH) after RI=2-3, when healthy fruits were used in the extraction process, but representing less than 30% decrease. However, this decrease is particularly important in the presence of pests and diseases.” 

Question: The results recorded in Table 2 do not show the reduction mentioned concerning bitterness. Please review this conclusion.

Ans: You are right. This reduction was only observed in 2020 and 2021 harvest years. This information was corrected as follows (lines 439-444):

“However, the presence of pests and fungal diseases may compromise the use of Cobrançosa oils in award-winning olive oil blends, mainly due to a decrease in the intensity of the pungent and bitter positive attributes, as observed in 2020 and 2021 harvests.”